# Strategies to Prevent Work Ability Decline and Support Retirement Transition in Workers with Intellectual and Developmental Disabilities

**DOI:** 10.3390/healthcare13141766

**Published:** 2025-07-21

**Authors:** Beatriz Sánchez, Francisco de Borja Jordán de Urríes, Miguel Ángel Verdugo, Carmen de Jesús Abena, Victoria Sanblás

**Affiliations:** University Institute on Community Integration (INICO), University of Salamanca, 37005 Salamanca, Spain; bea5sh@usal.es (B.S.); verdugo@usal.es (M.Á.V.); picky@usal.es (C.d.J.A.); victoriasanblascapi@hotmail.es (V.S.)

**Keywords:** active aging, retirement, work ability decline, disability, intellectual and developmental disabilities, transition to retirement

## Abstract

**Background/Objectives**: The aging of workers with intellectual and developmental disabilities is an emerging reality attributed to the rise in life expectancy and improved labor market access. In this study, “workers” is used as an inclusive, neutral term covering all individuals engaged in paid labor—whether employees, self-employed, freelancers, or those performing manual or non-manual tasks. It encompasses every form of work. It is crucial to comprehend the reality of aging workers from the perspectives of the primary individuals involved: the workers, their families, and supporting professionals. **Methods**: A qualitative study was developed, involving 12 focus groups and 107 participants, using NVivo 12 Pro for analysis; we used a phenomenological methodology and grounded theory. **Results**: A set of concrete needs was highlighted: among them, 33 were related to declining work ability due to aging and disability (WADAD), and 30 to transition to retirement. These needs were grouped into categories: workplace accommodations, coordination and collaboration, personal and family support, counseling and training, and other types of needs. **Conclusions**: This study establishes an empirical basis tailored to the needs of this group, enabling the development of prevention and intervention protocols that address WADAD and the transition to retirement.

## 1. Introduction

### 1.1. Intellectual Disability and Employment

Intellectual and developmental disability is characterized by limitations in intellectual functioning and adaptive behavior, affecting conceptual, social, and practical skills, and manifesting before the age of 22 years [1]. Diagnostic criteria include an IQ at least two standard deviations below the population mean and limitations in adaptive behaviors in at least one of three domains (conceptual, social, practical) or in the global score. In both cases, the standard error of measurement must be taken into account [1].

Employment is considered a universal and fundamental right for all people and represents a key goal in the process of transition to adult life. Article 27 of the Convention on the Rights of Persons with Disabilities [2] states that “States Parties recognize the right of persons with disabilities to work, on an equal basis with others; this includes the right to the opportunity to gain a living by work freely chosen or accepted in a labor market and work environment that is open, inclusive and accessible to persons with disabilities.” This is especially significant for people with intellectual and developmental disabilities, for whom work is the first step toward independent living and full participation in the community [3].

According to data from Spain published in December 2023 [4], the employment rate of people with disabilities in Spain is 27.88%, compared with 68.1% for people without disabilities. The unemployment rate is 21.4% compared with 8.6% among the population without disabilities. Within the working population with disabilities, 64.8% are between 45 and 64 years of age; this figure rises to 67.4% if we consider only those who are employed. Some authors have estimated that employment in Special Employment Centers (sheltered employment) in Spain accounted for 70% of employment contracts for people with disabilities in 2019 [5], and, more recently, 71% in 2022 [6]. However, Spain’s public employment service reports lower percentages: for example, in March 2024, out of 9361 contracts for people with disabilities, 5402 were made in Special Employment Centers, accounting for 57.7% [7].

Regarding supported employment in the ordinary (open) labor market, one study [8] made an approximation of the number of people hired in this type of employment in its latest available national report. However, this estimate is not complete as it is based only on data provided by entities that participated in the study. Consequently, Spain lacks reliable data in this regard due to the absence of a state registry.

Finally, with regard to the type of work carried out by people with intellectual disabilities, according to the EDAD 2020 survey [9], there are 20,166 people with intellectual disabilities working in Spain. Of these, 12,527 (62%) are engaged in elementary occupations. Due to sample limitations, it is not possible to provide reliable data on employment or job types by sector.

### 1.2. Declining Work Ability Through Aging and Disability and Transition to Retirement

Work ability decline through aging and disability (WADAD) is defined as the process of deterioration in work ability resulting from the interaction between disability and aging, which are two different but interrelated conditions [10]. This process carries the risk of a mismatch between a job role and the satisfactory performance of tasks, which may lead to forced retirement for reasons of health or job instability, causing a drastic change in lifestyle [10].

For any individual, the transition from paid employment to retirement is a significant milestone that impacts daily activity, emotional wellbeing, social relationships, and personal identity. Despite its importance, this reality remains under-researched in the context of intellectual and developmental disabilities [11,12]. To prevent retirement from being an abrupt change, workers with disabilities need a progressive and planned transition. This transition is acknowledged as an objective at the international level [13], and it should align with the principles of active aging, which aims to enhance quality of life in later years by optimizing health, participation, and security opportunities [14]. However, the effectiveness of this approach is frequently challenged due to the social exclusion traditionally experienced by people with disabilities [15].

In the European context, there is a dearth of scientific evidence addressing the specific needs of this group in relation to WADAD and the subsequent transition to retirement, primarily because these are emerging realities.

### 1.3. Active Participation in the Transition to Retirement

In the Spanish legislative framework [16,17], early retirement for people with intellectual disabilities is governed by two regulations and is defined by reduction coefficients and specific age and contribution period requirements. Royal Decree 1539/2003 establishes a minimum age of 52 years for people with a recognized degree of disability equal to or greater than 65%, and Royal Decree 370/2023 establishes a minimum age of 56 years for people with a recognized degree of disability equal to or greater than 45%.

People perceive aging differently, with some viewing it positively and others experiencing a loss of confidence and self-esteem [13]. However, active retirement carries significant benefits, including improved quality of life, enhanced physical and emotional wellbeing, increased motivation, and better self-image [18,19,20].

Pre-retirement planning, encompassing physical, leisure, and volunteering activities, along with continued support for education on retirement and aging, positively influences satisfaction levels [21,22,23].

Control over the timing of retirement is also crucial for a fulfilling aging experience [24]. Some people plan proactively and gradually, while others do so abruptly and reactively due to factors such as health concerns or job instability. Some may contemplate delaying retirement for financial reasons or for fear of inactivity, while others may consider partial retirement to combine work with other activities or to enjoy more rest [25]. What is needed are individualized, inclusive, and person-centered programs, as opposed to the many existing group-based and segregated approaches [26].

### 1.4. Planning the Transition to Retirement

Sometimes, older people with intellectual and developmental disabilities consider delaying retirement or taking partial retirement to combine work with other activities, enjoy more rest, manage finances, or alleviate the fear of inactivity [24].

The lack of access to day programs is a significant barrier for workers who have not previously engaged with such services, particularly for individuals in regular employment, who face inactivity post-retirement and may fear inadequate support [27]. That said, those in regular (open) employment generally have a more positive outlook on retirement, unlike their counterparts in sheltered employment who tend to associate retirement with boredom [28]. Insufficient funding for programs tailored to this group also represents a major barrier, compounded by low incomes resulting from retirement options [21]. As for health, inadequate access to health care and limited support for preventive care can contribute to unhealthy behaviors among people with intellectual and developmental disabilities [29].

Given their limited social networks, these individuals also require support for autonomy, especially after retirement [13]. It is therefore vital to nurture and strengthen these networks, as individuals with intellectual and developmental disabilities often favor work activity over retirement due to the significant value they place on social relationships in the workplace [27]. Although family support is common, it may diminish after retirement due to circumstances such as death, distance, or a lack of family interest [13]. Furthermore, the simultaneous aging of people with intellectual and developmental disabilities and their primary caregivers has an impact on social opportunities [30].

Accommodation post-retirement is also a major concern. The optimal situation would be for individuals to continue living with their family or in assisted housing, but in the medium to long term, nursing homes often become the only viable option, presenting a challenge due to a lack of staff training in disability care [13].

Financial security is key to choosing the appropriate time to retire [22]. Finally, in rural settings, activities and services for older retirees with intellectual and developmental disabilities are limited; however, the strength of the community may facilitate social support when compared with urban settings [31].

### 1.5. Purpose

The need for this study arises from the emerging and largely unexplored reality of WADAD and the subsequent transition to retirement. This necessity is further underscored by the increasing presence of people with intellectual disabilities in paid employment and nearing retirement age in Spain.

The purpose of this study is to identify a set of needs to effectively extend working life and facilitate the transition to active retirement for individuals with intellectual or developmental disabilities.

The ultimate aim of identifying these needs is to create two protocols: the first focused on prevention and intervention against WADAD, and the second centered on the transition to active retirement. In both protocols, the identified needs are accompanied by a number of action points.

## 2. Materials and Methods

A qualitative methodology using focus groups was chosen to explore the experiences, opinions, and perspectives of participants on specific issues [32]. Separate groups were formed for people with disabilities, family members, and professionals. This approach aligns with the suggestions of [33], which emphasize the importance of ensuring homogeneity in group profiles. However, the selection of participants may still vary.

The study proposal was approved by the Ethics Committee of the University Institute for Community Integration (INICO) of the University of Salamanca in 2020.

Twelve focus groups were held—six on WADAD prevention and six on retirement transition—with each block comprising two groups of family members, two groups of support professionals, and two groups of workers with intellectual and developmental disabilities.

We employed a process similar to that used in recent research on retirement and intellectual disability [11]. Adopting a phenomenological approach, we analyzed participants’ opinions about a specific reality and the needs associated with that reality, adhering to the framework outlined by [34]. We followed the steps of grounded theory [35,36] for data reduction and employed NVivo 12 Pro for data organization and transformation.

### 2.1. Participants

Participants were recruited by convenience using a nonprobability sampling method. The 12 groups comprised a total of 107 participants selected according to the following criteria:Workers with intellectual and developmental disabilities with a current employment contract, with verbal intelligibility and communication skills (n = 33; 30.84%).Family members of a worker with intellectual and developmental disabilities with a current employment contract (n = 27; 25.23%).Professionals working in aging or employment services (n = 47; 43.93%), comprising psychologists, job coaches, workshop teachers, or similar.

The majority were women (58.88%). Ages ranged from 26 to 81 years. In terms of educational attainment, 46.73% had been educated to university level, primarily consisting of professionals, while 34.58% had completed primary education, mainly comprising workers with intellectual and developmental disabilities. The majority of participants had either a generic intellectual and developmental disability (58.88%) or Down syndrome (36.45%). Finally, 49.53% of participant contributions pertained to regular (open) employment (whether supported or not), and 44.86% to sheltered employment (in Special Employment Centers). Further sociodemographic data are given in Table 1.

### 2.2. Procedure

Social organizations were contacted by email to secure their collaboration. They selected participants according to the provided guidelines to ensure diversity in age, years in employment, organization, sex, disability type, and employment type. The research team then reviewed these selections to achieve balance across all sociodemographic factors.

We collected participants’ sociodemographic data using a specially designed questionnaire. A checklist, developed based on topics considered relevant by the research team during the literature review, was used to guide the focus groups. The moderator introduced these topics through questions, allowing participants the freedom to share their perspectives openly.

The sessions lasted between 1.5 h and 2 h, and we obtained informed consent from participants beforehand, including permission record and process their data. For the workers with intellectual and developmental disabilities, the process was adapted using an easy-read preparation booklet, which they reviewed in advance with their support professionals. Additionally, we incorporated a break in the middle of the meeting to minimize fatigue, prevent distraction, and address any questions or concerns. Workers with intellectual disabilities and their family members received support from their designated professionals, as needed. Participants with sufficient autonomy joined remotely from home without any issues.

The focus groups addressing needs associated with the prevention of WADAD were conducted between February and March 2022. The focus groups addressing needs associated with the transition to retirement were conducted between February and March 2023.

The videoconference groups were recorded and then transcribed verbatim. Participants’ data were anonymized by assigning codes to prevent their identification during the research. The names of organizations, regions, and other individuals mentioned were also anonymized.

### 2.3. Data Analysis

For data reduction, we utilized codes (analysis categories) and sentiments expressed in the focus groups (whether positive or negative). The coding process, guided by grounded theory and employing three types of coding (open, axial, and selective), followed the constant comparison method [36,37]. This process required several iterations. If a previously unidentified code was deemed relevant, it was added. The team then reviewed earlier coding to determine whether the new code also applied to those segments.

After iterative coding, project maps were developed for each theme (WADAD prevention and retirement transition), merging literature-based and newly identified codes. These codes were defined in a codebook, and, in the final round, each quotation was systematically assigned to its corresponding code to streamline analysis. An interrater approach was used to achieve consensus and reduce subjectivity [38]. Additionally, focus groups with diverse profiles (professionals, family members, and workers with intellectual and developmental disabilities) enabled source triangulation, enhancing result verification and reliability [37].

## 3. Results

Table 2 depicts the categories that emerged from participants’ contributions during the group discussions. These categories are differentiated based on the frequency of their occurrence in relation to WADAD and the transition to retirement groups.

Additionally, the data are disaggregated for each participant profile, allowing us to observe the most frequently addressed topics overall and their importance within each group profile.

In the thematic area related to WADAD, the most recurrent topics were workplace adjustments and supports, totaling 110 references (37 from workers, 19 from family members, and 54 from professionals). Interinstitutional coordination and collaboration reached 87 references (8 from workers, 19 from family members, and 60 from professionals), while personal and family support accounted for 86 references (21 from workers, 33 from family members, and 32 from professionals). Other notable themes included counseling and training (64 references), legislation on retirement and disability (46 references), and prevention of abandonment, deterioration, or dismissal (46 references). Less frequently addressed topics were a lack of perceived needs (30 references), prevention of social network reduction (30 references), and financial security (26 references).

In the thematic area regarding the transition to retirement, the topics with the highest number of references were specific services for this process, with 283 mentions (67 from workers, 68 from family members, and 148 from professionals), active aging (261 references; 119 from workers, 81 from family members, and 61 from professionals), and personal and family support, with 243 references (102 from workers, 65 from family members, and 76 from professionals). Counseling and training received fewer mentions, totaling 161 references, followed by coordination and collaboration, with 113 references, and finally financial security, with 80 references.

In the context of work ability decline associated with aging and disability (WADAD), workers primarily focus on workplace adjustments and support (28.68%), as well as personal and family support (16.28%), and the prevention of job loss or dismissal (13.18%). Family members prioritize personal and family support (22.45%), coordination and collaboration among stakeholders (12.93%), and workplace adjustments (12.93%), indicating an interest in both close accompaniment and the proper organization and articulation of services. Professionals emphasize coordination and collaboration (24.10%), workplace adjustments (21.69%), and counseling and training (13.65%) reflecting a more structural and regulatory approach. Regarding the transition to retirement, workers highlight active aging (27.74%) and personal and family support (23.77%), along with retirement services (15.62%) and counseling and training (17.72%). Family members particularly emphasize retirement services (24.02%), active aging (28.62%), and personal and family support (22.97%), underscoring the importance of emotional and practical accompaniment. Professionals focus on retirement services (34.50%), personal and family support (17.72%), and coordination and collaboration (16.32%), with less emphasis on active aging, counselling and training, and financial security. Overall, the findings suggest that, while workers demand concrete supports and practical adaptations to maintain work ability, family members prioritize close support and the adequate organization of services, and professionals concentrate on coordination, regulation, and institutional management to facilitate both the extension of working life and a proper transition to retirement.

These results suggest that the effective management of work ability decline and the transition to retirement requires a balance between practical workplace supports, close personal support, and efficient coordination among the various stakeholders, while integrating the needs and perspectives of workers, family members, and professionals.

Delving deeper into the interrelationships across the categories identified in the groups, we can pinpoint a set of needs associated with WADAD (Table 3) and others linked to the transition to retirement (Table 4).

It is important to clarify the concept of parallel aging, which appears in both thematic blocks (need 19 in Table 3 and need 15 in Table 4). This term refers to the fact that, due to the earlier manifestation of the signs of aging in individuals with intellectual disabilities, a parallel aging process occurs between these individuals and their parents, who begin to experience aging simultaneously. This is due to the difference in the age of onset of aging between people with and without disabilities.

We present the most relevant needs in the following five subsections, illustrating them with representative quotations from participants.

### 3.1. Workplace Accommodations and Support Needs

To address and prevent WADAD, there is a need to adapt duties and make adjustments to job functions, while also incorporating a variety of tasks and rotations. This need is more achievable in sheltered employment than in regular (open) employment, as one professional explained:

“I understand that in the setting of a special employment center it’s perhaps easier to adapt tasks, to adjust a little, but in the mainstream setting, sometimes in companies, it’s more difficult”.(P.2.9)

Another crucial aspect involves shortening the working day and making adjustments related to duration. Through discussions in the focus groups, it became evident that this issue posed a challenge in calculating the required contribution period for early retirement. The challenge is more pronounced when there is an early onset of decline. However, recent legislative changes now permit partial working days to be counted as full contribution days. The contribution base is what decreases as remuneration is reduced. One worker with a disability described their experience:

In my company they do it bit by bit, they don’t go all of a sudden and make you retire, maybe they take away some hours, instead of 8 h they give you 6, then 4, and so on, bit by bit so that retirement is not very bad all of a sudden.(W.1.2)

Another need involves regulating public financial resources to counteract the decline in productivity. This includes assuming the financial cost of adjustments and supports, enabling individuals to continue working without their employers having to resort to reactive dismissals or permanent incapacity. Even with adjustments, the decline in productivity over time can become unsustainable, to the point that work incapacity is considered a viable way out for the worker, the company, and the family. One of the professionals put it this way:

In the SEC (special employment center), we’re obliged to have people who are employed but who are really doing occupational activities, at an occupational pace and output. Colleagues said that in regular employment this is obviously impossible and in a special employment center it’s unsustainable.… We can see that it’s not sustainable in the long term.(P.2.12)

### 3.2. Coordination and Collaboration Needs

At the internal level of the service provider organizations, there is a need for coordination with other services within the same entity. This collaboration is essential for joint planning with various types of professionals, and for the combination and implementation of other cross-cutting support measures. However, human resource shortages within the organizations and the growing financial burden on families to cover emerging support needs make this possibility difficult to realize. Furthermore, access to subsidized care resources is not permitted if the person has an income, which makes it difficult to combine reduced working hours with access to, for example, an occupational center. In the words of one professional, there is a need for:

“More flexibility, so that we could do something hybrid between the worker working and being able to use spaces in the occupational center, which isn’t legally possible”.(P.1.2)

A family member raised the following question:

It would be interesting to know if her organization does something to prepare them for retirement, but more than that, because she needs to have busy time and schedules, because she needs routine. To be busy doing things is very important, logically it’s not going to be a job, but you know, activities or things she might like, I don’t know if that’s offered.(F.1.1)

At the external level of the service provider organizations, there is a need for coordination with the other social and institutional stakeholders involved in providing workplace supports, whether directly or indirectly.

The life-course transition to retirement necessitates collaboration among support organizations, families, and employers. This collaborative effort enables the adjustment of expectations, the resolution of discrepancies, the identification of needs, the provision of supports, and the effective extension of working life. Moreover, it facilitates a smoother transition into retirement. In this regard, full-time jobs consistently yield more contributions in terms of time, and companies should take this factor into account in their offerings. One of the professionals made the following suggestion:

“I would train companies about this, about the aging of people with disabilities and the needs that are identified, then raise awareness within both the business and family environments”.(P.1.4)

In addition, fostering collaboration with other organizations within the disability sector and various public sector services is imperative. This involves simplifying procedures and enhancing cognitive accessibility in both employment and social security services. Furthermore, it is crucial to ensure that health services provide relevant information and support, facilitating the development of strategies to preserve the functioning capabilities of individuals with intellectual and developmental disabilities, with a particular emphasis on Down syndrome. This emphasis is essential due to the unique characteristics of aging in individuals with Down syndrome, such as a higher prevalence of Alzheimer’s disease and an earlier and more accelerated aging process. One professional shared the following perspective:

In both retirement and the general aging of people with Down syndrome and intellectual disabilities … in public health, we talk on a general level about a lack of knowledge about the aging processes of people with Down syndrome, and in many sectors they’re equated or normalized to the process of any other person, when in fact there are many studies that say it’s not the case.… And also there’s a lack of knowledge about the processes at a medical level for our users.… When it comes to giving sick leave or work incapacity issues we see that the health system isn’t up to date, there’s no specific training.(P.2.12)

### 3.3. Training and Counseling Needs

Of particular importance are job coaches and coworkers, who must be informed and trained regarding the implications of WADAD. As one professional pointed out, “I would ask for more coordination with public bodies, with community services” (P.1.5). Similarly, a worker had this to say:

First the organization, family, and coworkers.… You have to go where the informant is and they’ll tell you.… Let’s go where your coworkers are. We’ll get you together with them and you tell them that, OK? Your coworkers will have to know so they get a better idea, OK?(W.2.1)

This need for specific training extends to other professionals, who should acquire a better understanding of the signs of WADAD, its implications, and the relevant legal framework. Consideration should be given to the creation of participatory spaces for professionals to share experiences and knowledge. One professional made this point:

We find that when the first symptoms of aging appear, the main challenge is adapting the job to these new needs.… We need a lot of training to be able to adapt so that these people can continue with their working life.(P.2.10)

It is necessary to educate workers with intellectual and developmental disabilities and their families about retirement requirements, processes, possibilities, and compatibility with other pensions. This will support informed decision making and also alleviate the family’s concerns. As one worker requested:

Yes, I’d like to know about the paperwork to be completed and all that … that training is provided to everyone who’s also going to retire, to know where to find the positives and the negatives … the fears we have, what worries us and all that.(W.1.1)

The need for information on retirement extends to professionals, particularly concerning the prerequisites for eligibility and the aging processes in the context of disability. This pertains specifically to aspects such as legislation, tools, and strategies available within support organizations concerning aging, WADAD, and retirement. One professional highlighted the information gap:

We need research that we can see, to have objective and real data because the problem we’re starting with is that we don’t have data from earlier, for people who’ve been in these situations before, but we’re now encountering people in their forties and fifties who are working, and we don’t know what will happen in the future.(P.1.11)

### 3.4. Personal and Family Support Needs

For families, the challenge is to continue supporting their relative while simultaneously minimizing the stress caused by a new situation, even without a clear understanding of what is happening. As families play a key role, the difficulty of accepting aging and retirement in workers with intellectual and developmental disabilities can compromise successful collaboration. One family member had this to say:

I agree with my colleague, parents or relatives need information, but also training, on how we can help them take that step in life, because we haven’t thought about it. So we’re lost and so are they. We need someone to guide us so that we can make life easier for them.(F.1.2)

Another concern is the simultaneous aging of workers and their parents, as expressed by one worker, who worried that:

“if I’m old, I won’t have any family left”(W.2.4)

Similarly, a family member stated that:

“He still has some time left before he retires. But it’s definitely a concern, especially because we won’t be there when he retires. It’s better to have things prepared beforehand.(F.2.1)

When retirement becomes necessary and is feasible, there is a high degree of confusion around accepting it, navigating the associated aspects, and prioritizing a progressive and positive transition. One worker expressed the following concerns:

It worries me because I had been with my company for 16 years and, of course, I was useful working and so on. But when I’m not working and I’m starting to retire, I’m lost, because you have to do something to know that you’re worth something … not to say that you’re worth nothing. When you’re not working, well, I don’t know, I’m lost.(W.2.3)

There is a need for individualized support to help manage finances and to promote autonomy and independence, along with a psychological approach to address feelings of loneliness, while providing support for navigating retirement and other administrative and legal procedures. This need becomes more significant when family support is absent, and particularly so when procedures are carried out online with no guarantee of cognitive accessibility. One family member put it this way:

I imagine that what they’re thinking about and worrying about is the loss of the support they have from their parents.… I think that what worries my daughter the most is that her father or mother won’t be there, finding themselves alone or finding themselves a bit helpless, right.… They’re thinking about the loss of their parents and what will happen to them when we’re no longer around. I know that my daughter, when my mother died—and she was the grandmother she was closest to—that got her thinking about the situation with that bereavement.… What they’re thinking about is this, that at some point their parents might not be there, not about retirement itself.(F.1.4)

When the parents are still around and are also retired, it creates a dual impact. On the one hand, individuals with intellectual and developmental disabilities may be influenced or form expectations by observing their parents. On the other hand, there arises a need for support in managing family dynamics as a result of parents and children aging simultaneously. As one family member commented:

If they see us enjoying retirement, they’ll accept it. My daughter hasn’t thought about it.… But seeing her father and me enjoying retirement, I mean right now. Now we’re there more for her, so I don’t think it would be traumatic.(F.2.6)

### 3.5. Other Needs

There is a noticeable gap in the public provision of services for retiring individuals with disabilities. For many of them, this transition tends to occur earlier than for the general population, although this is not universally the case. The areas of social participation they once enjoyed are drastically reduced. The activities and resources available to them are not aligned with their actual age or their past life experience of engagement and participation in the community. As one professional urged:

“We have to offer them other services. Like, maybe going to the movies … meeting in bars for tapas, trips; in short, in my humble opinion I see a culture shock, I think there could be a culture shock”.(P.2.9)

The initial stages of the transition process, involving a reduction in working hours and gradual integration into other services, may be perceived as a threat rather than a support. This is because it can mean reduced contributions, and there are genuine fears about not meeting retirement requirements and the corresponding monetary benefits based on years of contribution. There is often a disconnect between the needs of the worker and the expectations of the family. One family member expressed the following concern:

I don’t really know anything about retirement or how many years they need to have worked or have been paying contributions or how many they have left, I have no idea about any of this really, it’s a whole new world.(F.1.1.)

The majority of the identified needs can be effectively met by creating individualized plans for the transition to retirement. These plans facilitate prior preparation, helping individuals comprehend and contextualize retirement. Additionally, they serve as a means to set expectations and promote self-determined choices. As one worker stated, “I’d like help preparing for … once I’m retired before, you know more or less how it works … what I’ll have left and so on, but I’d love help preparing for … not to be caught unawares”.(W.2.6)

The effective development of individual plans involves addressing the following key issues: the inadequate training of professionals and limited resources; low levels of contributions from workers; personal and family reluctance to reduce working hours; challenges in adapting job responsibilities, routines, or schedules, particularly in regular employment; difficulties in accessing and reconciling employment and care resources; workers’ reluctance to return to resources they relied on prior to being employed; and workers’ rejection of life choices that curtail their present levels of social participation.

## 4. Discussion

The results section highlighted some differences in the importance that different participant profiles give to the various issues identified. However, what is relevant is the complementarity of their points of view and the set of emerging needs collectively identified.

In this study, general strategies such as varying tasks, incorporating rotations, and shortening working hours were mentioned as potential accommodations. However, focus group data revealed a broader and more diverse range of workplace adjustments, including adaptations to work rhythms, the use of assistive technologies, ergonomic modifications, and the reinforcement of natural supports in the workplace. Likewise, it is important to acknowledge that different types of jobs—such as physically demanding versus sedentary roles—may require distinct types of adjustments. These issues are currently being addressed in the continuation of this research—the PROLAB APOYOS study—which aims to analyze in greater depth the types of supports provided, their sources, and their effectiveness in supporting workers with intellectual and developmental disabilities.

All of these findings are consistent with the literature, as other researchers [21,39] have already demonstrated that individuals who choose part-time work experience increased satisfaction during the transition to retirement and are able to engage in community activities while still being employed. Nevertheless, our study identifies issues concerning adjustments and productivity, which may drive individuals, particularly in mainstream employment, to consider work incapacity as a way out. This observation aligns with the findings of [21], highlighting the increased frequency of forced retirement in regular employment.

It is apparent that individuals with intellectual and developmental disabilities may, at times, want to sustain their working life because of what it brings them and make a gradual transition. Previous research has highlighted the benefits of a progressive transition [29,40] and the benefits of keeping active [41,42] on self-esteem, independence, physical and mental health [43], identity and security [41], community participation and social connectedness [30,40], cognitive decline [44,45], and a positive outlook on aging [40].

This research highlights the importance of coordinating internal and external services to provide comprehensive support for the prevention and management of WADAD, and for the transition to retirement. However, we have also pointed out that access to subsidized care resources is not permitted if the person has an income, which makes it difficult to combine reduced working hours with access to care resources, thus hindering a gradual transition. The family plays a key role, yet insufficient information often leads to resistance and a lack of acceptance. For their part, professionals emphasize the need for specific training. Workers with intellectual and developmental disabilities may experience concerns about retirement and therefore need access to appropriate information and support to better understand and evaluate these fears. Ref. [27] noted that retirement can be disruptive, especially when information about post-retirement activities is lacking. Ref. [42] highlighted that workers with intellectual and developmental disabilities often lack a complete understanding of retirement, resulting in feelings of dissatisfaction and a sense of diminished control. Similar sentiments are experienced by those who do comprehend the situation but cannot make decisions about it, supporting the observations of [28] regarding the insufficient preparation of individuals with disabilities for retirement decision-making processes.

This study addresses various financial issues that, according to the literature, are considered a primary concern. For example, ref. [21] underscores the critical role of financial aspects in retirement decisions, while [46] highlights how financial self-sufficiency enhances resilience against adversities in aging. In addition, the authors of [28] note that the decrease in income upon leaving the workforce is a barrier for people with disabilities when they retire. However, in our research, financial security was among the least frequently mentioned topics across participant groups.

Workers with intellectual and developmental disabilities are sometimes denied financial self-determination in their families, limiting them from pursuing the activities they desire [40]. This represents a challenge because, despite possessing financial resources, they may not have had the opportunity to learn how to manage them in preparation for the next stage of their lives as retirees. While it should be acknowledged that full financial autonomy may not always be achievable, it is crucial to consider who will provide this support once parents are no longer present.

Additionally, middle-aged individuals with disabilities face more limited social networks and fewer opportunities for relationships [40,47,48]. Our study also recognizes this reality as a barrier to be addressed during the transition to retirement.

Our study has revealed that WADAD and reduced productivity can sometimes result in sudden dismissals or recourse to solutions such as work incapacity, without proper transition planning. The transition to retirement can be either proactive and planned or reactive and sudden [49,50]. However, individuals with intellectual disabilities predominantly undergo reactive transitions, occurring abruptly and without sufficient planning [51].

This study underscores the fears and reservations of workers with intellectual and developmental disabilities and their families regarding post-retirement futures potentially filled with meaningless routines and tasks far removed from their employment experiences. These concerns align with findings from previous research [28,31,50].

The need for support both before and after retirement to facilitate a successful transition, alleviating fears and promoting continued social participation, has been demonstrated in other studies [21,27,48]. However, little is known about the types of supports or services needed by people with intellectual and developmental disabilities to foster active aging during the transition process [52].

Early planning is thus key to achieving a meaningful and positive transition. The authors of [30] previously emphasized the importance of promoting active aging through pre-retirement planning and preparation.

Active aging is considered the most effective response to the challenges posed by aging populations. However, there is substantial variation in how public bodies respond to the needs of older adults with intellectual and developmental disabilities. Other studies [12,42] reported that community-based programs were the most beneficial. While individuals with intellectual and developmental disabilities typically receive tailored supports, public initiatives to assist older adults with disabilities in their transition to retirement are limited [52]. Excluded from general social policies for older adults; this group requires specific services tailored to their needs [27]. Our research has identified a sense of dissatisfaction among individuals with intellectual and developmental disabilities and their families regarding general services for older people, which are designed for older people without disabilities. The same is true for residential services, complicating the prospect of aging in place. This difficulty has been reported by other authors [40,46,50], all of whom underscored aging in place as a fundamental pillar for aging individuals with intellectual and developmental disabilities.

### 4.1. Limitations

Conducting remote focus groups with workers nearing retirement or showing signs of decline presented challenges, as many participants were unfamiliar with the technology, and their support professionals were not present during the sessions. To facilitate participation, support technicians provided assistance before the sessions began. However, to ensure participants’ freedom of expression, the research team requested that these professionals leave the virtual room once the connection was established. During the focus groups, the research team provided timely support and, when necessary, contacted the technicians to intervene and assist participants. Additionally, easy-to-read manuals, breaks, and extra assistance were provided whenever possible.

In the family groups, difficulties arose due to a lack of volunteers and the advanced age of the participants, but these were overcome with the help of the collaborating organizations.

Although the wide age range of participants (from 26 to 81 years old) may be considered a limitation—especially given the study’s focus on the transition to retirement—this was an intentional decision. The research team chose to include a diverse sample in terms of age and profile in order to capture perspectives from individuals who were already beginning to consider retirement, as well as those who were not yet doing so. This diversity was deemed to enrich the findings by allowing the exploration of a broader variety of experiences, concerns, and needs related to aging in the workplace and preparing for retirement.

The translation of the participants’ statements may involve a loss of idiomatic meaning, which we consider acceptable and impossible to eliminate. To mitigate this issue, the translation was performed by a competent professional familiar with the area of study. The full texts of the original-language transcriptions are available in an open repository [53].

Finally, it is important to highlight that participant selection through collaborating organizations may introduce sampling bias. This limits the representativeness of the overall population of workers with intellectual and developmental disabilities and, consequently, the generalizability of the results to other contexts or groups. Additionally, given the qualitative nature and sample size, the findings should be interpreted with caution and considered as an exploratory approach rather than definitive conclusions that are applicable to the entire population.

### 4.2. Future Research Directions

Future research should focus on refining a robust and evidence-based tool that allows professionals to detect work ability decline, provide support through individualized plans, and progressively plan for the transition to retirement. The qualitative approach employed in this study could prove beneficial for developing protocols applicable to other types of disabilities (and for analyzing differences between groups, such as those in people with Down syndrome versus other intellectual disabilities). It enables the consolidation of knowledge derived from organizations with established practices in this field. Further, there is an opportunity to create a guide outlining common types of supports and adjustments for older workers with intellectual and developmental disabilities, along with guidelines to facilitate decision making.

## 5. Conclusions

This study aimed to identify the specific needs required to prevent work-related aging and decline among adults with disabilities (WADAD) and to support an active, planned transition to retirement for individuals with intellectual and developmental disabilities.

In this regard, there is a clear need to better understand the decline in work ability and how it progresses among aging individuals with intellectual and developmental disabilities. By conducting a qualitative analysis, this study examined the needs of this group, their families, and the professionals supporting them. The use of specific assessment tools could help professionals in the creation of individualized plans and supports to prevent and effectively manage WADAD, thereby enabling the self-determined extension of working life. Individual plans should be adjusted based on identified needs, encompassing task diversification and the integration of rotations to prevent boredom. Furthermore, these plans should align activities with workers’ capabilities, adjusting the pace and redistributing breaks accordingly. Similarly, changes in the job role should be instituted as needed, but in a controlled manner and with support, given the complexities associated with adapting to new responsibilities later in life. Vital aspects encompass ergonomic and technological support, a more flexible workday, and the training of coworkers who serve as natural supports within the workplace.

In this context, it is crucial to establish specialized prevention services. These services should address psychological aspects, offer prevention workshops, assemble interdisciplinary teams, improve the flow of communication between professionals within the organization, and foster connections with other social stakeholders. This comprehensive approach facilitates cohesive care across various areas of the lives of older workers with intellectual and developmental disabilities as they approach retirement. Additionally, training and information options on WADAD should be channeled to all stakeholders involved.

When the time to transition to retirement comes, individualized programs should consider workers’ preferences and provide detailed information on potential benefits and income sources. Support groups and programs addressing the pre-retirement stage can prove highly beneficial. Essential elements include providing information on leisure resources, cognitive exercises, and opportunities to enhance mental and physical health. Families also need advice and information to support them throughout the process. Additionally, legislation must be appropriately tailored to the identified needs.

Once the transition process is complete, there is a need for a comprehensive approach in the form of an active aging plan, encompassing health, leisure, social relationships, community participation, and housing options. Specialized training in this domain, along with collaboration between all public and private stakeholders in the areas of work, health, leisure, and independent living, is essential to address the complexities of aging in individuals with intellectual and developmental disabilities. This combination of specialized training and collaboration also serves to detect WADAD and facilitate the necessary adjustments for a fulfilling retirement.

These results offer relevant practical implications for policy design, service planning, and professional intervention. The identification of specific needs related to work ability decline and the transition to retirement provides a foundation for developing individualized, coordinated, and sustainable protocols. These should include workplace accommodations, early retirement planning, accessible training, and the development of community-based support networks. Thus, this study not only contributes empirical knowledge but also provides specific guidance to transform practice and improve the quality of life of people with intellectual and developmental disabilities as they age.

## Figures and Tables

**Table 1 healthcare-13-01766-t001:** Participant sociodemographic data.

		Workers * n = 33 (30.84%)	Family Members n = 27 (25.23%)	Professionals n = 47 (43.93%)	Total n = 107	%
Gender	Female	8	18	37	63	58.88
	Male	25	8	10	43	40.19
	Other/Not shared	0	1	0	1	0.93
Age	From	34	32	26	-	-
	To	63	81	58	-	-
Education	Primary	31	6	0	37	34.58
	Secondary	1	2	0	3	2.80
	Higher Secondary	0	5	2	7	6.54
	Vocational	1	4	5	10	9.35
	University	0	10	40	50	46.73
Etiology **	Intellectual and Developmental Disability	21	15	27	63	58.88
	Down Syndrome	11	9	19	39	36.45
	Autism Spectrum Disorder	1	1	1	3	2.80
	Other	0	2	0	2	1.87
Employment **	Open (supported or not)	14	15	24	53	49.53
	Sheltered	19	12	17	48	44.86
	Both	0	0	6	6	5.61

* With intellectual and developmental disabilities. ** Etiology of the intellectual disability of the participating worker, or in the case of family members or professionals, of the worker to whom they refer.

**Table 2 healthcare-13-01766-t002:** Number of quotes for the main topics of the focus group interviews (FGI), disaggregated by participant profile.

FGI	Topics	Workers *	Family Members	Professionals	Total No. of Quotes by Topic
Work Ability Decline through Aging and Disability (WADAD)	Adjustments and supports in the workplace	37 (28.68%)	19 (12.93%)	54 (21.69%)	110
	Coordination—collaboration	8 (6.20%)	19 (12.93%)	60 (24.10%)	87
	Personal and family support	21 (16.28%)	33 (22.45%)	32 (12.85%)	86
	Counseling and training	16 (12.40%)	14 (9.52%)	34 (13.65%)	64
	Legislation on retirement and disability	1 (0.78%)	17 (11.56%)	28 (11.24%)	46
	Preventing dropout, impairments, or dismissal	17 (13.18%)	12 (8.16%)	17 (6.83%)	46
	Needs not perceived	15 (11.63%)	7 (4.76%)	8 (3.21%)	30
	Preventing reduction of social networks	12 (9.30%)	16 (10.88%)	2 (0.80%)	30
	Financial security	2 (1.55%)	10 (6.80%)	14 (5.62%)	26
	No. of quotes by profile	129 (100%)	147 (100%)	249 (100%)	
Transition to retirement	Services for retirement	67 (15.62%)	68 (24.02%)	148 (34.50%)	283
	Active aging	119 (27.74%)	81 (28.62%)	61 (14.22%)	261
	Personal and family support	102 (23.77%)	65 (22.97%)	76 (17.72%)	243
	Counseling and training	76 (17.72%)	25 (8.83%)	60 (13.99%)	161
	Coordination—collaboration	19 (4.43%)	24 (8.48%)	70 (16.32%)	113
	Financial security	46 (10.73%)	20 (7.07%)	14 (3.26%)	80
	No. of quotes by profile	429 (100%)	283 (100%)	429 (100%)	

* With intellectual and developmental disabilities. **Note**: The percentages reflect the proportion of quotes for each topic relative to the total quotes for that participant profile within each thematic block. Therefore, they sum to 100% vertically (by participant profile), but not horizontally (across profiles). This presentation allows for a clearer comparison of the relative importance of topics within each profile, compensating for differences in the total number of quotes.

**Table 3 healthcare-13-01766-t003:** Needs identified in the WADAD focus group interviews, categorized by main topic.

WADAD Prevention and Intervention Needs	
Adjustments and supports in the workplace	1. Updating of on-the-job training 2. Creation of prevention services 3. Variation of tasks and positions 4. Ergonomic and technical support 5. Flexible working hours and reduction of working hours 6. Integration of labor and non-labor services 7. Peer support (avoiding overprotective attitudes) 8. Consider barriers such as rigidity of tasks (impossibility of job adaptations) and lack of resources to intensify support
Coordination and collaboration	9. Collaboration with other services of the entity to coordinate support 10. Adaptation of leisure services to work schedules 11. Implementation of mental agility programs, healthy habits, etc. 12. Collaboration with families to identify occupational impairment and extend support outside the workplace 13. Improvement of interprofessional communication with other areas (public and private), especially with the health sector 14. Coordination with social services for legal aspects of retirement, dependency, etc. 15. Collaboration with companies to create environments that are receptive to needs and accept adaptations in a favorable manner
Personal and family support	16. Family cooperation for work adaptations and to avoid overprotection 17. Family support to maintain worker’s health and autonomy 18. Financial difficulties to afford private employment services 19. Consideration of parallel aging and its implications 20. Emotional support for coping with the fear of family aging and other emotional support
Counseling and training	21. Advice on retirement legislation (family and professional) 22. Reducing family anxiety through counseling 23. Training professionals for early detection and intervention 24. Creation of forums on aging and disability 25. Psychological preparation of users for the aging process 26. Preparation for retirement, including formalities and financial aspects 27. Sensitization of companions/natural supports in early signs of decline
Other (legislation, financial stability, continuity of employment, etc.)	28. More flexible retirement requirements 29. Public funding for adaptation of positions 30. Prevention of job abandonment, disability, and reactive layoffs 31. Maintenance of social networks after retirement 32. Reducing working hours to balance leisure and work 33. Guarantee of security, financial stability, and housing

**Table 4 healthcare-13-01766-t004:** Needs identified in transition to retirement focus group interviews, categorized by main topic.

Transition to Retirement Needs	
Retirement services	1. Public services adjusted to older people with disabilities (generational and cultural gap due to early aging) 2. Collective programs for transition to retirement 3. Interdisciplinary teams (comprehensive approach to transition to retirement) 4. Support groups among retired and retiring workers 5. Development of programs and services to occupy leisure time with activities that promote physical and mental health 6. Individualized plans for transition to retirement, taking into account the needs, expectations, and fears of the workers
Coordination and collaboration	7. Support from companies to identify needs for prolonging working life 8. More full-time job offers to increase the possibility of reaching the required number of years of contribution 9. Collaboration between entity, family, and worker in retirement planning 10. Collaboration between entities to share practices and resources, because this is an emerging reality 11. Collaboration between health services and the disability entity to determine the health status of the employee
Personal and family support	12. Support in understanding and accepting the aging process and its consequences, as well as in adjusting to retirement 13. Support to family members in accepting decline and needs 14. Reconciliation between worker’s decisions and family’s expectations 15. Psychological support (to deal with parallel aging and family losses) 16. Assistance in financial and administrative retirement procedures 17. Autonomy, personal independence, and self-determination to decide 18. Identify community options tailored to individual needs
Counseling and training	19. Research on aging, retirement, and intellectual disability 20. Training for support technicians and family members in detecting signs of decline and legislation on retirement 21. Updating knowledge on aging and disability in the field of public health 22. Counseling and training for workers on retirement options and active aging 23. Training for companies in the recognition of signs of occupational impairment and support in the final phase of employment
Other (legislation, financial stability, active aging, etc.)	24. Reduction in working hours without affecting contributions for transition to retirement 25. Concerns about requirements for access to retirement (low contributions, etc.) 26. Information on compatible benefits and finding other sources of income to guarantee financial stability 27. Re-evaluation of degree of disability (possible increase to retire earlier) 28. Preventing and delaying cognitive decline and maintaining functionality 29. Routines and meaningful activities (leisure and training) after retirement 30. Social activities outside of work and community connection

## Data Availability

The transcript data are available in the University of Salamanca repository: Sánchez Herráez, B.; Jordán de Urríes Vega, F.B.; Verdugo Alonso, M.A.; Abena Abang, C.J.; Sanblás Capote, V. Transcripts from the Focus Groups of the PROLAB Project Research (Dataset); Gredos Repository, University of Salamanca: Salamanca, Spain, 2025. https://doi.org/10.71636/3j5p-9z03.

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
