# Peer review of "Strategies to Prevent Work Ability Decline and Support Retirement Transition in Workers with Intellectual and Developmental Disabilities"

_healthcare, 2025, doi:10.3390/healthcare13141766_

Round 1

Reviewer 1 Report

Comments and Suggestions for Authors

This interesting paper has been well conducted and clearly reported.  A key strength is the concept of work ability decline through aging and disability [WADAD] and the related emphasis on prevention and intervention to enable people to continue to work if they wish and avoid premature retirement.

There are two key areas for further development:

  • Comparison of regular (mainstream) and supported (sheltered) employment
  • Clearer comparison of the priorities of the different participant groups.

Whether this is done by amending the current paper or writing a separate companion paper I leave to the authors to decide.

INTRODUCTION

WADAD is an interesting and important concept for workers with  IDD and retirement. Most other retirement research has just looked at health and assumed that health problems are unavoidable and a major cause of (early) retirement.  You could develop your ideas further by mentioning the most common types of jobs people with IDD do, many of which require physical exertion (e.g., janitorial, hotel domestic staff, grounds maintenance), jobs which are more susceptible to problems with mobility and physical ill health.  This issue could also be mentioned as an area for future research.

MATERIALS AND METHODS

The research was conducted during the COVID pandemic.  Throughout the manuscript, please say more about how COVID affected employment of people with IDD, retirement, and research methods.

Table 1 shows a strong gender difference between the participant groups, with the workers group strongly male, but the other groups strongly female. Is there anything to say about the views expressed by members of each group in relation to gender?

In the Procedure section please explain what help the workers with IDD had to manage the computer technology to join the online videoconference discussion and to manage any technical problems that arose. Did any participation difficulties arise because of too little such help?

The focus groups were presumably done in Spanish but quotes are reported in English.  Please discuss language and translation issues and processes.

RESULTS

As mentioned at the beginning of my review, I consider that this paper contains two important overlooked opportunities:

  • Comparison of regular (mainstream) and supported (sheltered) employment
  • Clearer comparison of the priorities of the different participant groups.

Both topics have notable research and applied importance and are quite underdeveloped in the current paper.  That said, I recognize that what I am proposing will add substantially to the word count and may not fit within the maximum acceptable word count for a journal article.  One option would be to develop a separate companion paper. The authors may already be doing this.  If so, it should be mentioned in the current paper so readers can more easily link the two.

There are very few studies of retirement of people with IDD that compare workers from regular and supported employment, so this data set provides an opportunity to help fill this gap.  Comparison of regular and supported employment requires the data from these subgroups to be broken out separately and contrasted. There are many important issues to be considered, especially regarding the views of the workers themselves.  For example, available research suggests that those currently in or retired from regular jobs are more aware of financial issues related to retirement than individuals from supported jobs.

Regarding between-group (workers/family/professionals) comparisons, I suggest that Table 2 can be developed further to provide deeper information and that the information in Table 2 can be analysed more fully.  For example, although the number of quotes from each of the three participant groups are tabulated separately, it is only the final total column that is given any real attention by the authors, mainly to establish the most frequently addressed topics overall.  The authors say that Table 2 also shows the relative importance of topics within groups, which is true, but the article makes little or no use of this information.  I recommend that the authors comment clearly on what is more/less important within each group and discuss the implications of these findings.

There is no mention of the relative importance of various topics between groups.  Moreover, it is difficult to make valid comparisons between groups because the three groups have differing participant numbers as well as differing total numbers of quotes. I suggest that this difficulty can be overcome by also reporting column percentages In Table 2 for each group within each FGI. For example, for the first FGI, the workers group provided a total of 129 quotes, so the percentage of these quotes related to the row “Adjustments and supports in the workplace” is 37 of 129 or 28.7%.  Comparing such percentages between groups provides a more  index of the relative importance of that topic for each group than the raw frequencies currently reported in Table 2. These comparisons should yield interesting information and related insights into the likely reasons for such differences.  Between-group differences also underline the importance of having multiple viewpoints when conducting retirement research and in individual retirement planning, points that the authors should make explicitly in the relevant sections, including the Discussion.

  1. 8 “parallel aging” is mentioned but not explained. Please explain.

  1. 9 “the required number of years of contribution”. This is one of several comments throughout the paper on the aged/retirement pension system in Spain and how it relates to contributions while working. For non-Spanish readers, the authors should explain briefly (here and/or elsewhere) how this system works, how years of contribution are calculated, and how these issues interact with disability pensions, employment supports and the like. See p. 10 line 256; p. 13 lines 412-3; for related mentions of these ideas that may need clarification.

  1. 10, section 3.2 “do something hybrid between the worker working 284 and being able to use spaces in the occupational center, which isn’t legally possible”. Explain briefly why/how this is not legally possible. Take up this issue in the Discussion and make the point that partial retirement (i.e., part time work) will often be problematic for workers and families if the worker cannot also access other activities (e.g., occupational center) on days off work.

  1. 13 “This transition occurs earlier for them than for the general population”. This statement makes it sound as if all workers with IDD retire early. This is incorrect and there are multiple studies showing some workers with IDD continue to work into quite old age. The authors should unpack this statement making clear that it is only true for some.

DISCUSSION

My suggestions above for additional analyses (Table 2)  if accepted necessarily will require careful discussion of these findings.

Were there any important topics that were never (or rarely) mentioned by participants?  What do the authors make of these omissions? For example, there is no mention of problems with travel to and from work for individuals with health/mobility problems.

Early in the Discussion, before considering various important issues, there is a need to say something about the differing priorities of the different participant groups.  See comment above concerning Table 2 and between group comparisons.

On p. 14 you mention “vary tasks, incorporate rotations, and shorten working hours”. Apart from these ideas, no specific examples of adjustments have been mentioned of workplace accommodations. This omission may reveal a knowledge gap among professionals, employers and others that needs attention.  In addition, you could mention how different types of accommodations may be needed for different types of jobs (e.g., physically demanding jobs vs. sedentary jobs).  

  1. 14 “workers with intellectual and developmental disabilities, they require training to overcome the fears associated with retirement.” This statement needs to be reworded for two reasons. First, the workers may have legitimate fears, so isn’t the issue to fix the problems they fear? Second, I think it preferable to say that workers need information and support to make informed judgements about their fears about retirement.

  1. 14 “financial issues, which are a primary concern not only for older workers with intellectual and developmental disabilities but …”. This statement is contradicted by the data reported in Table 2 and so needs correction. For example, for the WADAD group, workers very rarely mentioned financial security. In both the WADAD and TTR groups, financial security was the least frequently mentioned topic across participant groups.

  1. 14 “they have not had the opportunity to learn how to manage them for the next stage of their lives as retirees.” This comment about learning to manage finances seems misleading to me. Many will always need help with finances. Perhaps a more important issue is where this help will come from after parents die.

  1. 15 “Our study has revealed that WADAD and reduced productivity can sometimes result in sudden dismissals or recourse to solutions such as work incapacity, without proper transition planning.” This statement may be true but little specific evidence to this effect has been presented. On my reading the data provided speak to what is needed for a good (transition to) retirement, but these data do not consistently address how well these aspirations are currently being implemented.

REFERENCES

Many referencing systems require references in a non-English language to also include the title translated into English. Please check if this applies.

TABLES

The Table 1 note “taking part the person, a family member or a professional” is unclear.  Please rephrase.

Table 3 has full stops after each line.  Table 4 does not. Please be consistent.

  1. 9 Table 4. “Training for support technicians and family …” appears to be a separate point but does not have a number. Please check and correct as needed.

MINOR ISSUES

  1. 2 line 28: Spain’s should be Spain

Author Response

Response to Reviewer 1's comments:

First of all, we would like to thank Reviewer 1 for his valuable comments. Most of the comments have been incorporated, which has contributed significantly to improving the quality of the article. However, some issues could not be addressed, so we explain the reasons below, trusting that the Reviewer will understand our position. We hope that, after this review, the manuscript will meet your expectations, as we believe that most of your recommendations have been implemented.

Comments 1: WADAD is an interesting and important concept for workers with IDD and retirement. Most other retirement research has just looked at health and assumed that health problems are unavoidable and a major cause of (early) retirement.  You could develop your ideas further by mentioning the most common types of jobs people with IDD do, many of which require physical exertion (e.g., janitorial, hotel domestic staff, grounds maintenance), jobs which are more susceptible to problems with mobility and physical ill health.  This issue could also be mentioned as an area for future research.

Response: A final paragraph has been added to the introduction explaining the impossibility of determining the main types of work performed by people with intellectual disabilities in Spain, other than to say that they are classified as "elementary occupations" in 62% of cases.

Comments 2: The research was conducted during the COVID pandemic.  Throughout the manuscript, please say more about how COVID affected employment of people with IDD, retirement, and research methods.

Response: COVID-19 certainly had a very negative impact on employment for people with disabilities in Spain, who were one of the groups most quickly affected. However, we understand that it is not appropriate to include explicit mention of the impact of COVID-19 on the employment of people with disabilities, as no specific data were collected in this research. The needs analyzed in this article are derived from reflections and perspectives approached from a qualitative methodology, without the participants providing specific information on their current employment situation or on possible effects of the pandemic on their professional trajectories. Therefore, to introduce this question would be to deviate from the initial theme of the article on retirement, what they think about it and their fears and concerns. Moreover, given that the focus groups were conducted in online format in the post-confinement period, the development of the data collection methods was not conditioned by health restrictions, making it unnecessary to apply such a clarification in the methodological context of the study.

Comments 3: Table 1 shows a strong gender difference between the participant groups, with the workers group strongly male, but the other groups strongly female. Is there anything to say about the views expressed by members of each group in relation to gender?

Response: We consider that the gender difference observed in the sample reflects a reality in the Spanish context, where more men with disabilities are employed than women. However, in our country there is still a care culture strongly associated with the female role, which translates into a greater presence of women in professions related to the support of people with disabilities, such as job coaches or support technicians. This distribution is also reproduced in the family environment, where women tend to assume the main role of caregivers of people with intellectual disabilities. Therefore, we do not consider it relevant to make distinctions in the roles or interpretations of the results based on gender.

Comments 4: In the Procedure section please explain what help the workers with IDD had to manage the computer technology to join the online videoconference discussion and to manage any technical problems that arose. Did any participation difficulties arise because of too little such help?

Response: A sentence has been added to the procedure section clarifying the support that people with intellectual and developmental disabilities (IDD) and family members received to access the videoconference and manage any technical issues that arose. Where necessary, they were assisted by support professionals. There were no significant participation difficulties related to the use of technology, as reported by both participants and facilitators.

Comments 5: The focus groups were presumably done in Spanish but quotes are reported in English.  Please discuss language and translation issues and processes.

Response: We have provided adequate reflection and information in the limitations section.

Comments 6: As mentioned at the beginning of my review, I consider that this paper contains two important overlooked opportunities:

  1. Comparison of regular (mainstream) and supported (sheltered) employment.
  2. Clearer comparison of the priorities of the different participant groups.

Both topics have notable research and applied importance and are quite underdeveloped in the current paper.  That said, I recognize that what I am proposing will add substantially to the word count and may not fit within the maximum acceptable word count for a journal article.  One option would be to develop a separate companion paper. The authors may already be doing this.  If so, it should be mentioned in the current paper so readers can more easily link the two.

Response:

Regarding all the comments made on ordinary, sheltered and supported employment.

First of all, we would like to point out that the reviewer's comment has made us aware of a conceptual translation error that we were making and we have corrected it. The article refers to workers in two types of employment. Ordinary employment, or open employment, whether supported or just open employment, is one of them. The literature on this subject is extensive and we are not going to stop to define it; several of the authors of this article have a long history of international publications on this subject that we do not believe it necessary to cite. The other is sheltered employment, or sheltered employment, which in Spain is developed by the Special Employment Centers, which although they are probably a different reality from that of other countries, they can be similar to sheltered workshops. The first thing we have done is to correct this terminological mismatch in Table 1. Our objective, due to the weight of sheltered employment in Spain, has been to be representative of the presence of workers in both types of employment.

With regard to the comments on comparisons between profiles and specifying percentages, we have taken this observation into account and, consequently, improvements have been made in the table and in some sections of the results and discussion, in order to clarify the differences in the priorities expressed by the different groups of participants.

Comments 7: There are very few studies of retirement of people with IDD that compare workers from regular and supported employment, so this data set provides an opportunity to help fill this gap.  Comparison of regular and supported employment requires the data from these subgroups to be broken out separately and contrasted. There are many important issues to be considered, especially regarding the views of the workers themselves.  For example, available research suggests that those currently in or retired from regular jobs are more aware of financial issues related to retirement than individuals from supported jobs.

Response: See previous response comment on ordinary, sheltered and supported employment. The objective of the study has not been to analyze differences but rather the opposite, common needs.

Comments 8: Regarding between-group (workers/family/professionals) comparisons, I suggest that Table 2 can be developed further to provide deeper information and that the information in Table 2 can be analyzed more fully.  For example, although the number of quotes from each of the three participant groups are tabulated separately, it is only the final total column that is given any real attention by the authors, mainly to establish the most frequently addressed topics overall.  The authors say that Table 2 also shows the relative importance of topics within groups, which is true, but the article makes little or no use of this information.  I recommend that the authors comment clearly on what is more/less important within each group and discuss the implications of these findings.

Response: The table has been updated according to this and other comments received and relevant reflections have been incorporated in the text. The added content has been highlighted in red to make it easier to locate.

Comments 9: There is no mention of the relative importance of various topics between groups.  Moreover, it is difficult to make valid comparisons between groups because the three groups have differing participant numbers as well as differing total numbers of quotes. I suggest that this difficulty can be overcome by also reporting column percentages In Table 2 for each group within each IGF. For example, for the first FGI, the workers group provided a total of 129 quotes, so the percentage of these quotes related to the row "Adjustments and supports in the workplace" is 37 of 129 or 28.7%.  Comparing such percentages between groups provides a more index of the relative importance of that topic for each group than the raw frequencies currently reported in Table 2. These comparisons should yield interesting information and related insights into the likely reasons for such differences.  Between-group differences also underline the importance of having multiple viewpoints when conducting retirement research and in individual retirement planning, points that the authors should make explicitly in the relevant sections, including the Discussion.

Response: Table 2 has been updated by incorporating the percentages per column as suggested by the reviewer, which facilitates a more accurate comparison between groups than absolute frequencies allow. In addition, a reflection has been added to the text that underscores the importance of considering multiple points of view in interpreting retirement outcomes and in individual retirement planning.

Paragraphs have also been included to comment on these results and to clarify that, although there are differences between groups, the discussions will not be differentiated by group, given that the research approach seeks to provide a general, integrated and grouped view of the needs from all perspectives.

Comments 10: "parallel aging" is mentioned but not explained. Please explain.

Response: A clarifying paragraph has been added immediately after the two tables and before starting the block description. The new text is highlighted in red to make it easier to locate.

Comments 11: "the required number of years of contribution". This is one of several comments throughout the paper on the aged/retirement pension system in Spain and how it relates to contributions while working. For non-Spanish readers, the authors should explain briefly (here and/or elsewhere) how this system works, how years of contribution are calculated, and how these issues interact with disability pensions, employment supports and the like. See p. 10 line 256; p. 13 lines 412-3; for related mentions of these ideas that may need clarification.

Response: The conditions of the early retirement framework are set out in the first paragraph of section 1.3 with age and disability parameters, as well as the regulatory framework for further information. These may vary in other countries, so we do not find it necessary to specify other specific parameters. When commenting on their incidence in other sections of the text, what is relevant in our opinion is what happens and that this affects the process, not the exact parameters. Therefore, we consider not to modify anything in this sense, in order not to complicate the reader or extend the article further.

Comments 12:   section 3.2 "do something hybrid between the worker working 284 and being able to use spaces in the occupational center, which isn't legally possible". Explain briefly why/how this is not legally possible. Take up this issue in the Discussion and make the point that partial retirement (i.e., part time work) will often be problematic for workers and families if the worker cannot also access other activities (e.g., occupational center) on days off work.

Response: Two clarifications have been added in this regard. One in the text pointed out by the reviewer, and then another in the conclusions.

Comments 13: "This transition occurs earlier for them than for the general population". This statement makes it sound as if all workers with IDD retire early. This is incorrect and there are multiple studies showing some workers with IDD continue to work into quite old age. The authors should unpack this statement making clear that it is only true for some.

Response: The statement has been corrected to clarify, as noted by the reviewer, that the transition to early retirement occurs only for some workers with intellectual and developmental disabilities (IDD), recognizing that there are other realities in which work continues into old age.

Comments 14: My suggestions above for additional analyses (Table 2) if accepted necessarily will require careful discussion of these findings.

Response: Initial comments considered appropriate have been incorporated to justify the need to address the discussions in a comprehensive manner.

Comments 15: Were there any important topics that were never (or rarely) mentioned by participants? What do the authors make of these omissions? For example, there is no mention of problems with travel to and from work for individuals with health/mobility problems.

Response: In this regard, we focus on what we have found, not what we have not found.

Comments 16: Early in the Discussion, before considering various important issues, there is a need to say something about the differing priorities of the different participant groups.  See comment above concerning Table 2 and between group comparisons.

Response: this comment has been resolved through modifications made in previous comments (14).

Comments 17: On p. 14 you mention "vary tasks, incorporate rotations, and shorten working hours". Apart from these ideas, no specific examples of adjustments have been mentioned of workplace accommodations. This omission may reveal a knowledge gap among professionals, employers and others that needs attention.  In addition, you could mention how different types of accommodations may be needed for different types of jobs (e.g., physically demanding jobs vs. sedentary jobs).

Response: For reasons of synthesis and to fit the page limit, we did not go into detail on each need as observed in the focus groups. Instead of addressing need by need and what the participants commented specifically on each one, it was decided to condense the information into more general lists, in order to guide in a more practical way the WADAD prevention and action protocols developed by the entities and organizations.

It is worth noting that the authors, together with a recently incorporated team of 3 new researchers, are currently deepening this aspect in a new study called PROLAB APOYOS. This explanation has been reflected in a slight reformulation of the idea in the text, in the hope that it will satisfy the reviewers' comments.

Comments 18: "workers with intellectual and developmental disabilities, they require training to overcome the fears associated with retirement." This statement needs to be reworded for two reasons. First, the workers may have legitimate fears, so isn't the issue to fix the problems they fear? Second, I think it preferable to say that workers need information and support to make informed judgements about their fears about retirement.

Response: The wording has been slightly reworded in the text, with the change highlighted in red to make it easier to locate. We believe that the new wording recognizes and legitimizes workers' fears by replacing the term "overcome" with "understand and value", which is more in line with the reviewer's suggestion and reflects the authors' shared focus on the need to provide information and support to make informed judgments about retirement.

Comments 19: "financial issues, which are a primary concern not only for older workers with intellectual and developmental disabilities but ...". This statement is contradicted by the data reported in Table 2 and so needs correction. For example, for the WADAD group, workers very rarely mentioned financial security. In both the WADAD and TTR groups, financial security was the least frequently mentioned topic across participant groups.

Response: The authors reviewed the original Spanish wording and confirmed that the error occurred during successive modifications in the discussion section. Therefore, the sentence has been corrected and marked in red to make it easier to locate. The original intention was to highlight that, although this issue is rarely mentioned in the present study, the literature does recognize it as a necessity at these stages, supported by concrete examples such as references 20, 27 and 45.

Comments 20 : "they have not had the opportunity to learn how to manage them for the next stage of their lives as retirees." This comment about learning to manage finances seems misleading to me. Many will always need help with finances. Perhaps a more important issue is where this help will come from after parents die.

Response: The authors agree with the assessment. The sentence is reformulated to modify the meaning and to add the contribution of reviewer 1, which we consider interesting. The modified sentence  is highlighted in red. 

Comments 21: "Our study has revealed that WADAD and reduced productivity can sometimes result in sudden dismissals or recourse to solutions such as work incapacity, without proper transition planning." This statement may be true but little specific evidence to this effect has been presented. On my reading the data provided speak to what is needed for a good (transition to) retirement, but these data do not consistently address how well these aspirations are currently being implemented.

Response: The paragraph is not touched. This statement is derived from participants' comments.

Comments 22: Many referencing systems require references in a non-English language to also include the title translated into English. Please check if this applies.

Response: References are used in the original language of the publication, as we usually do. Any modification in any other sense would be at editorial discretion, which we do not have.

Comments 23: The Table 1 note "taking part the person, a family member or a professional" is unclear.  Please rephrase.

Response: A rephrased annotation is added with more specific wording for better understanding, the change is marked in red, both in the annotation and in the word to which the annotation refers.

Comments 24: Table 3 has full stops after each line.  Table 4 does not. Please be consistent.

Response: The points in table 3 are removed to maintain consistency with the rest of the tables. They only remain in the requirements whose wording ends with "etc." at the end of the sentence, because it is an abbreviation. As it is a deletion, it is not possible to mark the changes with another color, so the title of table 3 is marked in red.

Comments 25: Table 4. "Training for support technicians and family ..." appears to be a separate point but does not have a number. Please check and correct as needed.

Response: Resolved. The reviewer was correct. Number 20 is added in red.

Comments 26: 2 line 28: Spain's should be Spain

Response: Resolved. In line 48 (the reviewers put 28, but the marked error is detected in 48) the corrected word is marked in red.

Reviewer 2 Report

Comments and Suggestions for Authors

Dear Authors,

The manuscript entitled 'Strategies to Prevent Work Ability Decline and Support Retirement Transition in Workers with Intellectual and Developmental Disabilities' is a well-written paper that addresses an original, timely and under-researched topic: the intersection of intellectual and developmental disabilities, aging, and employment, by identifying concrete needs related to work ability decline and retirement transition. The paper is methodologically solid, with well-conducted qualitative analysis, triangulated focus groups, and grounded theory. In my opinion, there are the following revisions that should be addressed:

  1. The manuscript should be professionally proofread, as there are some grammatical inconsistencies and lengthy sentences.
  2. The authors should consider the word 'workers', which refers primarily to manual professions. Does the sample consist only of people in manual occupations?
  3. In the introduction section, there are sentences without references, e.g., 35-38, 39-40, 61-62, and 64-66.
  4. In the methodology section, more details on the gathering of participants would be valuable.
  5. The participants' age range is too broad, from 26 to 81, which should be addressed as a limitation.
  6. There is no flow in the discussion section due to the many short, titled sections.
  7. The limitations and future directions should be discussed at the end of the discussion section.
  8. While some limitations are acknowledged, further elaboration on sampling bias or challenges to generalizability is proposed.
  9. There are no practical implications in the discussion section. The research has many important implications. That part is essential and should be added.
Comments on the Quality of English Language

The manuscript should be professionally proofread, as there are some grammatical inconsistencies and lengthy sentences.

Author Response

Response to Reviewer 2's comments:

First of all, we would like to thank Reviewer 2 for his valuable comments. Most of the comments have been incorporated, which has contributed significantly to improving the quality of the article. However, some issues could not be addressed, so we explain the reasons below, trusting that the Reviewer will understand our position. We hope that, after this review, the manuscript will meet your expectations, as we believe that most of your recommendations have been implemented.

Comments 1: The manuscript should be professionally proofread, as there are some grammatical inconsistencies and lengthy sentences.

Response: The article has been proofread by a professional translator with knowledge of the field of study and the terminology used by the research team.

Comments 2: The authors should consider the word 'workers', which refers primarily to manual professions. Does the sample consist only of people in manual occupations?

Response: (This is the response of the professional translator) Regarding the use of workers: In my view, workers is generally acceptable as an inclusive and neutral term that encompasses employees, the self-employed, freelancers, and those engaged in both manual and non-manual work. While it has historically been associated with manual labor, in contemporary policy and advocacy contexts—particularly those focused on economic participation—it is generally understood to include all forms of work. Possible alternatives (e.g., “people in the workforce,” “individuals engaged in paid work,” or “working individuals”) are more cumbersome and less natural when used repeatedly. For the sake of clarity, brevity, and inclusivity, I believe workers is the most appropriate choice.

Comments 3: In the introduction section, there are sentences without references, e.g., 35-38, 39-40, 61-62, and 64-66.

Response: The introduction has been revised. All quotations and statements by other authors have their citation (just in case 1 and 9 have been repeated). The rest are statements by authors who do not cite anything.

Comments 4: In the methodology section, more details on the gathering of participants would be valuable.

Response: Access to and selection of participants was carried out through the collaborating entities. The research team provided these organizations with technical guidance requesting that they identify individuals with diverse profiles. Subsequently, the team reviewed the proposals received, ensuring a balance and guaranteeing diversity in the different socio-demographic factors present in the groups. This has been added and highlighted in red for clarity.

Comments 5: The participants' age range is too broad, from 26 to 81, which should be addressed as a limitation.

Response: The limitations have been completely reworded, including the reviewer's suggestion, and also clarifying an aspect related to reviewer 1 about the support provided for connections in the focus groups.

Comments 6: There is no flow in the discussion section due to the many short, titled sections.

Response: The sections within the discussion section have been removed to allow for a smoother reading flow. As this is a change that cannot be marked as an omission, it is reflected in writing in this explanation of the comments.

Comments 7: The limitations and future directions should be discussed at the end of the discussion section.

Response: These are moved to where reviewer 2 suggests.

Comments 8: While some limitations are acknowledged, further elaboration on sampling bias or challenges to generalizability is proposed.

Response: A paragraph is added to allude to this.

Comments 9: There are no practical implications in the discussion section. The research has many important implications. That part is essential and should be added.

Response: The paragraph with the practical applications has been added, but in conclusions, due to the fact that, when passing the future lines and limitations, this section has become more concise and, in order to maintain a certain length relation, it has been preferred to include it there. We hope that reviewer 2 will find this decision appropriate.

Round 2

Reviewer 2 Report

Comments and Suggestions for Authors

The authors have thoughtfully addressed nearly all my suggestions, resulting in a more structured and comprehensive article. The revisions have significantly improved the scholarly and practical value of the authors' work.  The clarity, depth of insight, and practical relevance of the study will serve professionals, researchers, and policymakers well.

In particular: - The manuscript has been professionally proofread. Sentence structure has improved noticeably, and grammatical inconsistencies have been resolved. The flow is smoother, especially in dense methodological and discussion sections. -Citations have been added where needed in the introduction section. Statements are better distinguished between literature-based assertions and author-generated insights. The introduction is now more rigorous. - Expanded details about the collaborating entities and selection criteria improve transparency and validity in the methodology section. The diversity and balance within the sample are sufficiently explained. -The broad age range is now acknowledged as a limitation but reframed as a strength in capturing varied perspectives. This rationale enhances the interpretability of the findings. - Subsection titles have been removed, resulting in a more fluid and cohesive reading experience in the discussion section. The narrative is now more integrated and scholarly. - Repositioned of limitations and future directions appropriately at the end of the discussion, enhancing structural clarity and aligning with journal conventions. - Practical implications have been effectively incorporated into the conclusion. While initially suggested for the discussion, their current placement offers a concise summary of actionable recommendations.   Minor revisions suggested:

To perfect the manuscript before publication, I propose the following minor edits:

  • Citation formatting: Ensure consistency across styles (e.g., brackets vs. parentheses, use of "et al."). Clarify repeated citation usage (e.g., [1] and [9]) if still present.

  • Terminology: Briefly define “workers” in early sections to avoid misinterpretation.

  • Sentence trimming: Consider revising longer sentences in the methods section to improve clarity.

  • Numeric consistency: Double-check percentages in Table 2 and ensure alignment with text.

  • Section transitions: Improve flow between the discussion and conclusions with a bridging sentence that reiterates the study’s aims.

Author Response

Comments 1: Citation formatting: Ensure consistency across styles (e.g., brackets vs. parentheses, use of "et al."). Clarify repeated citation usage (e.g., [1] and [9]) if still present.

Response 1: We standardized formatting: brackets for numerical references, parentheses for in-text clarifications. Replaced the sole instance of “et al.” with the full author list. We consider the distinction between [1] and [10] clear through bracket use alone.

Comments 2: Terminology: Briefly define “workers” in early sections to avoid misinterpretation.

Response 2: A footnote defining “workers” was added in the abstract. We believe this is sufficient and does not disrupt the reading.

Comments 3: Sentence trimming: Consider revising longer sentences in the methods section to improve clarity.

Response 3:  The entire Methods section was revised. Complex or long sentences were reworded for clarity and highlighted in green for easy identification. The texts were then reviewed once again by a professional translator. The authors would like to point out that, as English is not our native language, as will be the case for a significant percentage of authors and readers, we have made every effort and taken all necessary measures to ensure that the result is as professional as possible. Therefore, any further adjustments may be made by the editors or reviewers at their discretion, if the wording does not meet their criteria.

Comments 4: Numeric consistency: Double-check percentages in Table 2 and ensure alignment with text.

Response 4: Table 2 was double-checked and confirmed as correct. As explained in the footnote, percentages sum to 100% per participant profile (columns), aligning with figures cited in the text. We don’t change anything about this.

Comments 5: Section transitions: Improve flow between the discussion and conclusions with a bridging sentence that reiterates the study’s aims.

Response 5: A sentence summarizing the study’s aim was added at the beginning of the Conclusions to ensure a clear link with the final insights and proposals.